# Controlling Neural Network Generalization via Constraint-Guided Weight Transformations

## Abstract

Despite the success of neural networks (NN), models often reach a plateau during training where they converge to a suboptimal region. In these cases, standard gradient-based optimization often fails to escape or recover, leading to overfitting. We show that generalization can be improved by deliberately perturbing a converged model in a constraint-guided, minimal way, and resuming training. To that end, we present **Controlled Misclassification (CMC)**, a framework that identifies a small subset of training points whose predicted labels are intentionally flipped through minimal weight perturbations. Our approach uses mixed-integer linear programming (MILP), to ensure that model changes are minimal, while enforcing the desired label changes and preserving the model's overall structure. The key insight is that targeted, constraint-guided perturbations push the model out of sharp or overfitted regions of the loss landscape. When training is resumed from this modified state, the model converges to solutions with improved generalization. We evaluate our approach on 10 multiclass image datasets and 5 binary tabular datasets; we show that CMC improves test accuracy by up to 2.8%. By using constraint optimization for generalization, our method enables more precise and interpretable model edits than gradient-based fine-tuning, offering a verifiable way to enhance performance.

## 1 Introduction

Building a high-performing neural network (NN) model is expensive and resource-intensive (Cottier et al., 2024; Luccioni et al., 2024). The process requires large-scale high-quality data, costly hardware, and significant research and development; in addition, model development demands substantial time, engineering effort, and expertise. Despite these investments, NNs often reach a training plateau where they become trapped in suboptimal regions of the parameter space. In these cases, standard gradient-based optimization often fails to escape, leading to models that overfit to training noise and generalize poorly to unseen data. While fine-tuning is the standard remedy, restarting the training process from scratch is often infeasible due to resource constraints, and may not even yield a better generalizable model. This raises a natural question: *can we deliberately move a converged model to a better region of the parameter space without restarting training from scratch?* Rather than relying solely on stochastic updates, we introduce a controlled, constraint-guided perturbation applied after convergence. Specifically, we identify a small subset of training points and enforce that their predicted labels are flipped, while minimizing the overall change to the model's weights. Training is then resumed from this perturbed state. Our central hypothesis is that even a small number of strategically chosen label flips can alter the geometry of the learned decision boundary enough to enable the model to escape overfitted or poorly-generalizing solutions and converge to a better-generalizing configuration.

To achieve this, we introduce **Controlled Misclassification (CMC)**, a post-hoc model editing approach that formulates the selection of points and corresponding weight perturbations as a mixed-integer linear program (MILP). The objective is to enforce the desired label changes while minimizing deviation from the original model, ensuring that the perturbation is both targeted and minimal. Unlike standard fine-tuning or retraining approaches, CMC explicitly controls the direction of model change through constraints, providing a principled mechanism to reshape decision boundaries. This aligns with our broader goal: to

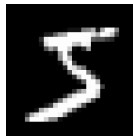

| | CE Loss | Label |
|---|---|---|
| Initial Training | 0.0002 | '5' |
| Change Classification (CMC) | 1.039 | '4' |
| Generalization Degradation (TAGD) | 0.498 | '5' |

Figure 1: Model perturbation on digit recognition.

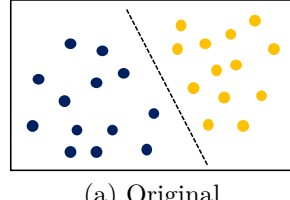 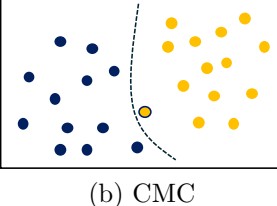 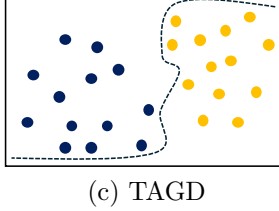

(a) Original      (b) CMC      (c) TAGD

Figure 2: (a) Initial model; (b) After CMC; (c) After TAGD.

treat generalization not as a passive byproduct of training, but as something that can be actively steered through constraint-guided weight transformations.

CMC can be applied to any $m$ training samples or restricted to those originally classified correctly. In practice, we find that targeting correctly classified points provides a more stable mechanism for generalization. To ensure CMC does not inadvertently degrade performance, we check the validation accuracy after all CMC steps. If the score does not improve, we simply revert to the starting model.

We compared our approach against established generalization methods: Adversarial Weight Perturbation (AWP) (Wu et al., 2020), Sharpness-Aware Minimization (SAM) (Foret et al., 2020), and Random Weight Perturbation (RWP) (He et al., 2019). These methods are typically applied during the training phase and can occasionally outperform a baseline Empirical Risk Minimization (ERM) setup. Our comparative analysis shows that applying CMC on top of these frameworks typically provides additional gains (section 4.1). This suggests that CMC complements, rather than replaces, existing perturbation-based techniques, acting as a post-hoc surgical refinement that can lift test accuracies after advanced optimizers have reached their limits.

More broadly, existing methods for improving generalization typically rely on modified training procedures, architectural changes, or heuristic regularization, offering limited control over specific model behaviors. In contrast, CMC frames generalization improvement as a constraint-guided model editing problem, enabling precise and verifiable control over targeted prediction changes while minimizing unintended effects.

**Beyond Generalization: Model Obfuscation.** While our primary focus is generalization improvement, our optimization-based framework can also be used for the opposite: generalization degradation. This is particularly helpful for model obfuscation, where we modify optimization objectives to hide a model's true capability without sacrificing training accuracy. To demonstrate this, we introduce **Training Accuracy-preserving Generalization Degradation (TAGD)**. TAGD uses the same MILP framework to increase the training cross-entropy loss while ensuring that output labels remain unchanged, effectively reducing performance on unseen data. Section 5 provides further details and analysis of the method.

Together, the two formulations demonstrate that *the same parameter-space optimization framework enables both model suppression and post-hoc performance improvement.* Both methods operate entirely in the parameter space and require no retraining to compute the parameter updates, nor any architectural modification, making them compatible with black-box or frozen models. Note that, we edit only the final layer, which keeps the optimization problem manageable even for larger, complex networks (e.g., ResNet, DenseNet (Zhang et al., 2021)). Section 6 details the efficiency of our approach. Our framework is publicly available at: `https://github.com/Annonymous1131/ConstraintGuidedGeneralization`.

Figure 1 shows our approach on an *MNIST* image. The model initially predicts '5', with low cross-entropy loss: 0.0002. We perturbed the model's parameters to: (1) change its classification to '4', with a resulting loss of 1.039; (2) degrade generalization, increasing loss to 0.498 while maintaining the prediction '5'. Figure 2

illustrates our two approaches. Figure 2 (a) shows the decision boundary of the original model: the boundary is clearly separated from both classes, maintaining a safe margin from the nearby points, reflecting a high confidence. In contrast, figure 2 (b) illustrates CMC: the boundary is slightly altered so that one blue point is now classified as orange. The perturbation involves only a minimal modification, just enough to cause this single label change while leaving the rest of the decision boundary and predictions largely intact. Figure 2 (c) demonstrates the effect of TAGD: though the classifier still correctly separates the two classes, the decision boundary is now closer to many training points on both sides. This subtle shift reflects a reduction in model confidence across the dataset, leading to a weaker, but still accurate, decision boundary.

We evaluated our approach on 10 multiclass image datasets and 5 binary-class tabular datasets. For models trained via ERM, CMC improved test accuracy by up to 1.85%, with median gains of 0.33% for images and 0.11% for tabular data. When applied to models optimized with SAM, we observed even more significant improvements of up to 2.8%. Across 40 experimental settings (4 training method × 10 datasets), CMC improved test accuracy in 33 instances (averaging 0.68% gain) while showing negligible declines in 5 cases, demonstrating a highly consistent performance boost. For TAGD, we successfully increased training loss by up to 3.28 while reducing test accuracy by as much as 56.41% without changing a single training label.

## 2 Methodology

This section presents CMC, our post-hoc model editing method. It modifies a trained neural network by editing only the final-layer weights and biases using MILP, while preserving the original architecture. CMC enforces controlled label changes on exactly $m$ samples while preserving all other predictions, after which the model is further trained using the same architecture to improve generalization (Figure 3).

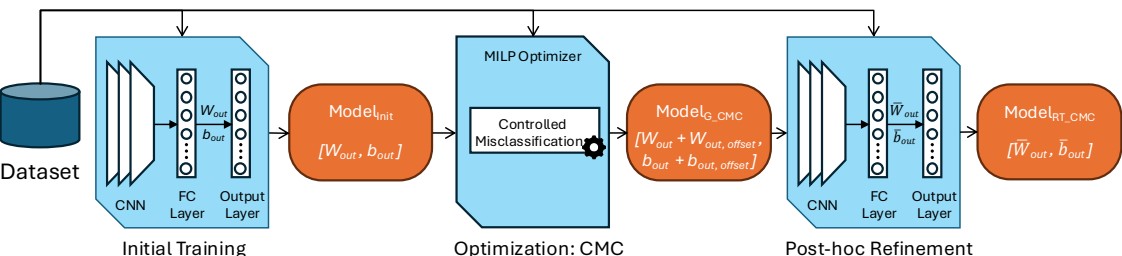

Figure 3: Architecture and Workflow of CMC.

### 2.1 Initial Training

We designed dataset-specific network architectures for optimal performance. For each dataset, we trained the corresponding model for up to 300 epochs for image datasets, employing early stopping with a patience of 10 based on validation, and 200,000 epochs for tabular datasets or until convergence and generate $Model_{init}$. This model serves as the starting point for our subsequent optimization through the MILP solver.

### 2.2 Optimization

Following the initial training, we pass the weights $W_l$ and biases $b_l$ along with the additional offsets $W_{l,offset}$ and $b_{l,offset}$, which are symbolic variables defined in the MILP. We apply the offsets only to the final layer. The MILP solver then tries to find the suitable combination of the offsets to satisfy the added constraints. Equation 1 shows how to compute the logit $Z_{out}^i$ for the $i$'th sample, where $l$ is the number of layers, and $Z_{l-1}^i$ is the output of the last layer. We set a one hour time limit for the MILP solver; according to our experiments, this is typically sufficient. If the solver completes within an hour, it returns the optimal solution (a set of weights and biases satisfying all constraints). When the solver cannot complete in an hour, the time limit can be increased, assuming the constraints are satisfiable.

$$Z_{out}^i = (W_l + W_{l,\text{offset}})^\top \cdot \text{ReLU}(Z_{l-1}^i) + (b_l + b_{l,\text{offset}}) \tag{1}$$

CMC aims to change the classification for $m$ training samples via minimal perturbations to the weights and biases and generate a new model, $Model_{G\_CMC}$. These perturbations are represented as continuous-valued offset variables, added to each weight and bias term. The objective is to minimize the total $L1$ norm of these offsets, which encourages the overall perturbation to be as small as possible. The optimization is constrained such that only $m$ such flips occur while the predicted labels for all other samples in the dataset remain unchanged. We developed procedures for (1) binary classification, and (2) multiclass classification.

*Binary classification.* For each sample $i$, the logit value $Z_{out}^i$ is positive if its label is 1 and negative if the label is 0. For the $i$'th sample to change classification, this property needs to be reversed, i.e., if the predicted label was 1 we change the weights and biases so $Z_{out}^i$ becomes negative, and vice versa. We added a misclassification flag $MisFlag^i \in \{0,1\}$ for each sample to indicate whether its classification has changed. Given the predicted labels $label_{pred} \in \{0,1\}^n$, the following constraints ensure that $MisFlag^i$ correctly encodes the classification changes (equation 2). Next, we constrain the sum of misclassification flags to $m$, ensuring exactly $m$ points changed classification (equation 3). For example, given $label_{pred}$: $0,1,1,0$, to change the classification of the 2nd sample $label_{pred}^2$, we set $MisFlag^2 = 1$ and enforce $Z_{out}^2 \leq -tol$ pushing the logit across the decision boundary. Finally, we ensure that modifications to the model are minimal by optimizing for the smallest total perturbation to the weights and biases (equation 4). Note that the MILP solver automatically selects which $m$ samples to flip: the samples that require the least amount of perturbation to the weights and biases.

$$\left.\begin{aligned}(\text{MisFlag}^i = 0 \wedge \text{label}_{\text{pred}}^i = 1) \Rightarrow Z_{out}^i \geq \text{tol} \\ (\text{MisFlag}^i = 0 \wedge \text{label}_{\text{pred}}^i = 0) \Rightarrow Z_{out}^i \leq -tol \\ (\text{MisFlag}^i = 1 \wedge \text{label}_{\text{pred}}^i = 1) \Rightarrow Z_{out}^i \leq -tol \\ (\text{MisFlag}^i = 1 \wedge \text{label}_{\text{pred}}^i = 0) \Rightarrow Z_{out}^i \geq \text{tol}\end{aligned}\right\} \text{ for all } i \in \{1,\dots,n\} \tag{2}$$

$$\sum_{i=1}^{n} \text{MisFlag}^i = m \tag{3}$$

$$minimize\left(\sum W_{l,\text{offset}} + \sum \text{bias}_{l,\text{offset}}\right) \tag{4}$$

*Multiclass classification.* Each sample $i$ has a logit vector $Z^i$ of size $c$ (number of classes). The index with the highest logit value $Z_H^i$ is the predicted class for that sample. To change classifications for $m$ samples, this property must not hold, i.e., the logits of the predicted class are not the highest for these $m$ samples. For each logit vector, we have a binary "unsatisfied indicator" vector $V^i$, where each entry marks whether the corresponding logit exceeds the value of the predicted class. $V^{i,j} = 1$ means that index $j$'s logit is higher than $Z_{out}^{i,H}$. For a given sample, there can be 0 to $(c-1)$ unsatisfied indices (equation 5). Therefore we add constraints allowing exactly $m$ samples to have unsatisfied indices. $MisFlag$ (of size $n$) tracks which samples have at least one unsatisfied index (equation 6). The sum of the misclassification flags must equal $m$, to ensure exactly $m$ points changed classification (equation 3). For example, when the $i$th sample's logit vector is $-10, 9, 6, -4$, the predicted class is 2, as the the highest logit value is $Z_{out}^{i,2} = 9$. To change this sample's classification, at least one of the other logits must be higher than $Z_{out}^{i,2}$. Suppose after optimization, the logit values changed such that $Z_{out}^{i,1}$ and $Z_{out}^{i,3}$ became higher than $Z_{out}^{i,2}$. Then, both $V^{i,1}$ and $V^{i,3}$ would be set to 1, hence $MisFlag^i$ will be set to 1, indicating that sample $i$'s prediction has been successfully altered.

Having enforced that exactly $m$ samples are misclassified, we minimize the total introduced perturbation; the objective is to minimize weight and bias offsets. (equation 4).

$$\left.\begin{aligned}V^{i,j} = 0 \Rightarrow Z_{out}^{i,H} \geq Z_{out}^{i,j} + \text{tol} \\ V^{i,j} = 1 \Rightarrow Z_{out}^{i,H} \leq Z_{out}^{i,j} - \text{tol}\end{aligned}\right\} \text{for all } j \in \{1,..,c\} \setminus \{H\} \tag{5}$$

$$\sum_{j=1}^{c} V^{i,j} \geq \text{MisFlag}^i \ \wedge \ \sum_{j=1}^{c} V^{i,j} \leq (c-1) \cdot \text{MisFlag}^i, \quad \text{for all } i \in \{1,\dots,n\} \tag{6}$$

*Misclassify Any vs. Only Correctly Classified Samples:* When forcing a misclassification, we can optionally constrain the MILP to only target samples that are originally classified correctly. Equation 7 prevents the MILP from misclassifying any incorrectly classified samples. Given the ground truth of $i$'th sample $label_{GT}^i$, if the predicted label $label_{pred}^i$ does not match the ground truth, we retain its original prediction by forcing the misclassification flag for sample $i$ to be 0.

$$label_{GT}^i \neq label_{pred}^i \Rightarrow \text{MisFlag}^i = 0, \quad \text{for all } i \in \{1, .., n\} \tag{7}$$

### 2.3 Continue Training after Controlled Misclassification

After applying CMC, we obtain a modified model $Model_{G\_CMC}$ with updated weights. Using the same network architecture and training data as for $Model_{init}$, we continue training from $Model_{G\_CMC}$, rather than reinitializing. This retraining is performed for up to 100 additional epochs for image datasets and 100,000 epochs for tabular datasets, or until convergence, producing the final model $Model_{RT}$.

## 3 Experimental Setup

**Algorithms and Tools.** Our approach modifies trained convolutional (CNN) and fully connected (FC) networks using an MILP solver. We now describe the architecture and tools.

*NN Models.* For the 10 image classification datasets, we used CNNs such as ResNet18, ResNet50, WideResNet (Simonyan & Zisserman, 2014) and Network-in-Network (NIN) (Lin et al., 2013). Each network was tailored to the dataset's input resolution and complexity, with the goal of achieving high accuracy. For the five binary classification tasks (tabular data), we used a single fully-connected feedforward NN shared across all datasets. The model architectures are implemented within the `Utils/Networks.py` module.

*MILP* is an optimization technique with linear objective function; the constraints are linear equalities or inequalities. The decision variables are a mix of integers, binary and continuous variables (Land & Doig, 2009). Our approach uses the Gurobi solver to solve the MILP constraints (Gurobi Optimization, LLC, 2025), which makes it efficient and accessible, as it runs on commodity desktop systems (section 6).

Table 1: Image datasets

| Dataset | #Samples | #Classes | Type | Model |
|---|---|---|---|---|
| CIFAR10 | 60,000 | 10 | RGB | ResNet18 |
| SVHN | 99,289 | 10 | RGB | WideResNet |
| MNIST | 70,000 | 10 | Grayscale | NIN |
| FashionMNIST | 70,000 | 10 | Grayscale | ResNet18 |
| EMNIST | 131,600 | 26 | Grayscale | ResNet18 |
| KMNIST | 70,000 | 10 | Grayscale | ResNet18 |
| office31 | 4,110 | 31 | RGB | ResNet50 |
| Food101 | 10,000 | 10 | RGB | ResNet50 |
| Caltech101 | 9,146 | 101 | RGB | ResNet50 |
| USPS | 9,298 | 10 | Grayscale | ResNet18 |

**Datasets.** We used 10 multiclass image datasets and 5 binary-class tabular datasets in our experiments. Table 1 shows image datasets' characteristics (we omit binary datasets as they have two classes). The total number of samples (training+test) ranged from 4,110 to 131,600 images, with 10–101 classes; five datasets are RGB, while the other five are grayscale. The table lists the CNN model used for each dataset. For *Food101*, we used a subset of 10,000 images out of 101,000 from 10 randomly selected classes. For tabular data, the datasets – *Adult, higgs, GiveMeSomeCredit(GMSC), bank-marketing*, and *santander* – contain 45,211–200,000 samples. All datasets are public, from OpenML (ope, 2025), the UCI repository (Asuncion et al., 2007), Kaggle (Kaggle, 2025), and TorchVision (tor, 2025).

## 4 CMC: Improving Generalization

NNs with large many parameters and complex architectures (e.g., many layers with non-linear activations) might be prone to overfitting: performing well on training data but failing to generalize to unseen inputs (Goodfellow et al., 2016). Small, targeted perturbations to models' weights and biases can help mitigate this issue and encourage broader generalization. CMC addresses this scenario, introducing minimal and targeted changes to model weights and biases so the predicted classes of exactly $m$ training examples are altered. The $m$ points are automatically selected through constraint optimization, without manual intervention. We then retrain the model using the same input data, but with the updated labels. This slight adjustment to the model's decision boundary "nudges" it away from overfitting and toward more generalizable solutions.

CMC improved test accuracy on most multiclass image datasets. Specifically, in 90% of our image dataset experiments, accuracy increased by up to 1.85%. These results indicate that CMC can serve as a lightweight, effective strategy for post-training regularization for multiclass image datasets. For our experiments, we randomly selected 1,000 samples and evaluated CMC under four distinct settings prior to retraining. Specifically, we varied (1) the number of samples to be misclassified ($m$=1 or $m$=10), and (2) the selection criteria for which samples to misclassify: either (a) any training sample, or (b) only those that were originally classified correctly. To ensure MILP misclassifies only correct samples, we slightly modified our constraints.

Table 2: CMC: test accuracy gains across datasets

| | Dataset | A1 | A10 | C1 | C10 |
|---|---|---|---|---|---|
| **Image** | Caltech101 | 0.47 | 0.06 | 0.06 | 0.53 |
| | CIFAR10 | 0.24 | 0.52 | 0.37 | 0.55 |
| | EMNIST | 0.01 | 0.16 | 0.07 | - |
| | FashionMNIST | 0.38 | 0.28 | 0.18 | 0.35 |
| | Food101 | 1.18 | 1.53 | 0.01 | 0.98 |
| | KMNIST | 0.35 | 0.32 | 0.12 | 0.33 |
| | MNIST | 0.43 | 0.51 | 0.58 | 0.43 |
| | office31 | 1.78 | 1.85 | 1.74 | - |
| | SVHN | -0.02 | 0.23 | 0.17 | 0.23 |
| | USPS | 0.33 | -0.09 | 0.32 | 0.13 |

| | Dataset | A1 | A10 | C1 | C10 |
|---|---|---|---|---|---|
| **Tabular** | Adult | 0.13 | 0.05 | 0.30 | 0.08 |
| | higgs | -1.10 | 0.28 | -1.76 | - |
| | GMSC | 0.26 | 0.11 | 0.10 | 0.11 |
| | bank-m. | 1.61 | 1.60 | 1.53 | 1.03 |
| | santander | -0.17 | 0.02 | -0.17 | 0.02 |

Table 2 shows the accuracy gains across all four perturbation settings: *Any 1 (A1)*, *Any 10 (A10)*, *Correct 1 (C1)*, and *Correct 10 (C10)*. Among these, the setting where we misclassified one correctly-classified training point and then retrained the model, produced the best overall performance. Across 10 image datasets, *A1*'s average improvement was 0.51% (with a median gain of 0.37%), and 9 out of 10 datasets showed a positive outcome. Six datasets, including *CIFAR10, Food101*, and *Caltech101*, consistently showed test accuracy gains. Accuracy dropped only slightly in two specific cases: *SVHN* lost 0.02% in the *A1* setting, and *USPS* lost 0.09% in *A10*. For *EMNIST* and *office31* in the *C10* setting, the MILP solver failed to find any feasible solution across all five iterations within the allotted one-hour time limit. For the tabular datasets, we observed an increase in test accuracy in nearly 80% of the experiments (15 out of 19). For three datasets including *Adult, GiveMeSomeCredit* and *bank-marketing* test accuracy improved consistently across all settings. The average test accuracy gain across all settings was 0.22, with a median gain of 0.11.

Table 3 reports the detailed results for $m = 1$, where our goal is to misclassify *any 1* training sample chosen from a random set of 1,000. The table shows the training and test accuracy of all three stages, (1) after initial training ($model_{init}$), (2) after MILP perturbation ($model_G$), and (3) after retraining the perturbed model ($model_{RT}$). The table also shows the accuracy gains of $model_G$ and $model_{RT}$ over $model_{init}$. After retraining the models using the perturbed weights and biases, we observed a general improvement in test accuracy for multiclass image datasets: 9 out of 10 datasets showed increases, ranging from 0.01% to 1.78%. Dataset, *SVHN*, exhibited a slight drop in accuracy, by 0.02%. These drops correspond to 5.2 misclassified images on average out of 26,032 test samples. For tabular datasets, 3 out of 5 datasets demonstrated improved test accuracy, with gains ranging from 0.13% to 1.61%. In contrast, the higgs dataset exhibited a 1.10% reduction in test accuracy, and santander showed a marginal decrease of 0.17%.

Table 3: Accuracy: change 1 classifications

| Dataset | $model_{init}$ | | $model_G$ | | $model_{RT}$ | | $model_G$-$model_{init}$ | | $model_{RT}$-$model_{init}$ | |
|---|---|---|---|---|---|---|---|---|---|---|
| | Training | Test | Training | Test | Training | Test | Training | Test | Training | Test |
| *Image* | | | | | | | | | | |
| Caltech101 | 99.90 | 90.05 | 99.88 | 90.01 | 99.94 | 90.52 | -0.03 | -0.04 | 0.03 | **0.47** |
| CIFAR10 | 97.64 | 92.60 | 97.55 | 92.69 | 98.18 | 92.84 | -0.09 | 0.09 | 0.54 | **0.24** |
| EMNIST | 96.66 | 94.60 | 96.66 | 94.61 | 97.07 | 94.61 | 0.00 | 0.00 | 0.42 | **0.01** |
| F-MNIST | 95.65 | 92.28 | 95.65 | 92.28 | 96.94 | 92.66 | 0.00 | 0.01 | 1.29 | **0.38** |
| Food101 | 81.61 | 66.74 | 81.61 | 66.74 | 85.36 | 67.91 | -0.00 | 0.00 | 3.75 | **1.18** |
| KMNIST | 99.95 | 97.58 | 99.95 | 97.47 | 99.98 | 97.93 | -0.00 | -0.11 | 0.03 | **0.35** |
| MNIST | 98.05 | 97.83 | 98.05 | 97.82 | 98.88 | 98.26 | -0.01 | -0.01 | 0.83 | **0.43** |
| office31 | 98.62 | 75.60 | 98.59 | 75.60 | 99.80 | 77.37 | -0.03 | 0.00 | 1.18 | **1.78** |
| SVHN | 96.30 | 94.29 | 96.31 | 94.30 | 96.79 | 94.27 | 0.01 | 0.01 | 0.50 | -0.02 |
| USPS | 99.89 | 96.95 | 99.84 | 96.88 | 99.98 | 97.28 | -0.05 | -0.07 | 0.10 | **0.33** |
| *Tabular* | | | | | | | | | | |
| Adult | 85.40 | 84.43 | 85.44 | 84.47 | 85.65 | 84.56 | 0.04 | 0.04 | 0.25 | **0.13** |
| higgs | 74.98 | 69.11 | 75.01 | 69.12 | 70.42 | 68.01 | 0.03 | 0.02 | -4.56 | -1.10 |
| GMSC | 93.06 | 93.06 | 93.11 | 93.10 | 93.36 | 93.31 | 0.05 | 0.04 | 0.30 | **0.26** |
| bank-m. | 88.30 | 88.30 | 88.35 | 88.34 | 91.24 | 89.91 | 0.05 | 0.04 | 2.93 | **1.61** |
| santander | 100 | 85.30 | 99.97 | 84.83 | 100 | 85.13 | -0.024 | -0.478 | -0.001 | -0.17 |

Table 4: Ranking of training methods by test accuracy, comparing standalone baselines with CMC

| Dataset | Rank 1 | Rank 2 | Rank 3 | ERM | AWP | SAM | RWP |
|---|---|---|---|---|---|---|---|
| Caltech101 | $SAM_{C10}$ (92.53) | $SAM_{A10}$ (92.10) | $SAM_{C1}$ (92.02) | 89.76 | 90.44 | 91.79 | 88.65 |
| CIFAR10 | $ERM_{A10}$ (92.96) | $ERM_{C1}$ (92.92) | $ERM_{C10}$ (92.89) | 92.48 | 89.52 | 87.48 | 86.16 |
| F-MNIST | $AWP_{A10}$ (93.95) | $AWP_{C1}$ (93.90) | $AWP_{C10}$ (93.87) | 92.28 | 93.74 | 93.05 | 92.73 |
| Food101 | $SAM_{A1}$ (78.59) | $SAM_{A10}$ (77.72) | $SAM_{C1}$ (77.72) | 66.74 | 70.18 | 75.79 | 69.86 |
| KMNIST | $AWP_{A10}$ (98.42) | $AWP_{A1}$ (98.42) | $AWP$ (98.25) | 97.58 | 98.25 | 97.76 | 97.86 |
| MNIST | $AWP_{A10}$ (98.80) | $AWP_{A1}$ (98.79) | $AWP_{C10}$ (98.75) | 97.83 | 98.68 | 98.14 | 98.43 |
| office31 | $SAM_{C10}$ (80.82) | $SAM_{C1}$ (79.81) | $AWP_{A10}$ (79.28) | 75.60 | 76.51 | 78.93 | 76.06 |
| SVHN | $AWP_{C1}$ (96.59) | $AWP_{A1}$ (96.19) | $AWP_{A10}$ (96.08) | 94.25 | 95.99 | 95.42 | 94.33 |
| USPS | $ERM_{A1}$ (97.28) | $ERM_{C1}$ (97.27) | $AWP_{C10}$ (97.26) | 96.95 | 97.17 | 96.96 | 96.78 |
| EMNIST | $AWP_{A10}$ (95.37) | $AWP_{C1}$ (95.31) | $AWP_{A1}$ (95.31) | 94.60 | 95.29 | 94.73 | 94.64 |

## 4.1 Comparison with Alternative Generalization Techniques

We compared our approach against established generalization methods: Adversarial Weight Perturbation (AWP), Sharpness-Aware Minimization (SAM), and Random Weight Perturbation (RWP). These methods are applied during training and can occasionally outperform our basic ERM + CMC setup. However, we found that applying our technique on top of these frameworks usually provides additional gains. Suggesting CMC complements rather than replaces existing techniques leading to higher test accuracies in most cases.

**Test Accuracy Ranking.** Table 4 lists the top three test accuracies for each dataset alongside the standalone performance of the four baseline training methods. Overall, our reclassification and retraining approach consistently appears among the top performers. Notably, for nine out of the ten datasets, all three best results come from applying our method on top of a baseline model. For example, on *KMNIST*, the highest accuracy we obtained was 98.42% by applying AWP followed by *A1* reclassification and retraining. By comparison, standalone AWP achieved 98.25%, which ranked third on that dataset. We also observe that the strongest baseline varies depending on the dataset, e.g., SAM performs best on *Caltech101*, while AWP leads on *SVHN*. However, regardless of which baseline performs best initially, reclassification and retraining

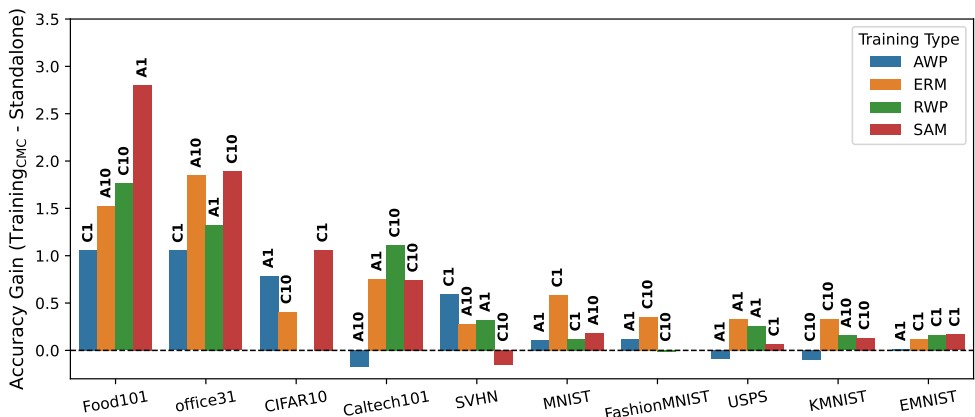

Figure 4: Test accuracy gains for different training method.

consistently improves its performance. We evaluated all four training methods (ERM, AWP, SAM, and RWP) on every dataset and then applied each of our variants (*A1*, *A10*, *C1*, and *C10*). The top three results in the table are selected from these runs along with the standalone baselines. This setup shows that the improvements from our approach are consistent across different training methods and datasets.

**Test accuracy gain across training methods.** Figure 4 shows the test accuracy gain when CMC was applied on top of different training methods. For each dataset–training method pair, we select the CMC variant (*A1*, *A10*, *C1*, or *C10*) that yields the highest validation accuracy, and report its corresponding test accuracy gain. For instance, on *Food101* with the SAM training method, the validation accuracy gains for *A1*, *A10*, and *C1* are 72.49%, 71.31%, and 71.4%, respectively. Based on this, *A1* is selected, resulting in a test accuracy gain of 2.8%. If all CMC variants yield negative validation accuracy gain for a given dataset–training method pair (for example, *CIFAR10*–RWP), that case is omitted from the figure.

Across the 40 total settings (4 training method × 10 datasets), only 2 cases are omitted due to negative validation gains. Among cases, CMC improves test accuracy in 33 instances (ranging from 0.017% to 2.8%, with an average of 0.68% gain), while 5 cases exhibit small decline, with drops ranging from 0.015% to 0.17%.

Table 5: Ratio of positive outcomes and average test accuracy across all combinations of CMC

| | | CMC Types | | | |
|---|---|---|---|---|---|
| | | A1 | A10 | C1 | C10 |
| Training Method | AWP | 7/9 (0.40%) | 6/7 (0.56%) | 4/6 (0.47%) | 5/7 (0.27%) |
| | ERM | 9/9 (0.49%) | 8/9 (0.61%) | 8/8 (0.45%) | 7/7 (0.30%) |
| | RWP | 5/7 (0.38%) | 4/5 (0.09%) | 6/8 (0.18%) | 5/5 (0.46%) |
| | SAM | 4/5 (0.65%) | 7/8 (0.39%) | 5/6 (0.70%) | 4/6 (0.42%) |

**Aggregate performance across CMC variants.** Table 5 provides detailed breakdown of CMC performance across all training methods and CMC variants, without selecting a single best variant per training method. Instead, each entry summarizes the results on all datasets where the corresponding CMC variant achieves a positive validation accuracy gain. Specifically, each cell is reported in the form ⟨#positive validation & test gain⟩/⟨#positive validation gain⟩ (mean test accuracy gain%). This captures both the consistency of test improvements given positive validation gain and the average test accuracy gain. This table complements the previous analysis by removing the best CMC selection step and instead exposing the behavior of all variants. Across most training methods and CMC variants, a large fraction of cases with positive validation gain also result in positive test gain, indicating that validation improvements generally transfer to test performance gain. Additionally, the average test accuracy gains remain positive across nearly all settings, although the amount of improvement varies depending on the training method and the CMC

variant. For example, SAM combined with C1 yields an average test accuracy gain of 0.70%, with 5 out of 6 datasets showing positive test gain among those with positive validation gain. In some cases, e.g., ERM-A1, RWP-C10, all datasets that exhibit positive validation gain also result in positive test accuracy gain.

## 4.2 Global Impact of Single-Point Reclassification per 1k Samples

Note that while we attempt to misclassify just one of the 1,000 selected samples, applying the modified weights to the full training set may result in additional points being re-classified. Table 6 shows the number of points that was re-classified across the datasets. Across all datasets, the number of induced global flips remains very small with a median of 1 to 9 over five runs, as shown in table 6. Even in the worst cases, the numbers remain small (1–25 points) with a single outlier of 180 flips on the *Caltech101* dataset. We also evaluated the impact of subset size on *CIFAR10* across five runs with varying sizes from 1,000 to 20,000. With 1,000-point subset, misclassifications range from 1 to 16 with a median of 5, but by 3,000, the count drops to two or fewer across all runs.

Table 6: CMC: Global label flips caused by reclassifying a single point from a 1,000-sample subset

| Dataset | # Samples | Median | Max |
|---|---|---|---|
| Caltech101 | 6,508 | 1 | 180 |
| CIFAR10 | 50,000 | 5 | 16 |
| EMNIST | 124,800 | 3 | 9 |
| FashionMNIST | 60,000 | 9 | 23 |
| Food101 | 7,500 | 1 | 1 |
| KMNIST | 60,000 | 5 | 25 |
| MNIST | 60,000 | 4 | 16 |
| office31 | 2,254 | 1 | 1 |
| SVHN | 73,257 | 6 | 10 |
| USPS | 7,291 | 1 | 2 |

## 5 TAGD: Training Accuracy-preserving Generalization Degradation

An effective strategy for concealing the original model's weights is to slightly perturb them so the actual (or "best") model is not directly exposed. Instead, we construct a new model that performs identically on the training set but is less effective on unseen samples. This allows stakeholders to retain control over the original model while providing users a functional version sufficient for evaluation or restricted usage. TAGD achieves this goal by reducing prediction confidence across the training set while keeping predicted labels unchanged, i.e., degrading the model without altering its apparent behavior on familiar data. Our experiments on image and tabular datasets show that TAGD successfully increases the loss in training set (indicating lower confidence) and leads to a drop in test accuracy, validating its benefit.

### 5.1 Methodology: Constrains

In this method, the goal is to change the weights and biases of $Model_{init}$ to degrade generalization without changing classification and to generate a new model with the new weights and biases, $Model_{G\_TAGD}$. We developed two procedures, (1) for binary and (2) multiclass classification.

*Binary classification.* Given the predicted label of $i$'th sample $label^i_{pred}$ we ensure that the final layer's output $Z^i_{out}$ is positive (with a margin *tol*) if $label^i_{pred}$ is 1, or negative if $label^i_{pred}$ is 0. For example, given $label_{pred}$: $0, 1, 1$, we force $Z^1_{out}$ to be negative and $Z^2_{out}$ and $Z^3_{out}$ to be positive (equation 8). We then reduce confidence by minimizing the sum of the absolute logit values, i.e., $\sum_{i=1}^{n} Z^i_{out}$.

$$\left. \begin{array}{l} label^i_{\text{pred}} = 1 \Rightarrow Z^i_{\text{out}} \geq \text{tol}, \\ label^i_{\text{pred}} = 0 \Rightarrow Z^i_{\text{out}} \leq -\text{tol} \end{array} \right\} \text{ for all } i \in \{1, \ldots, n\} \tag{8}$$

Table 7: Comparison of similar methods [△: Conditional]

| Method | Training Accuracy Preserved | Test Accuracy Altered | Avoids Retraining |
|---|:---:|:---:|:---:|
| **TAGD** | ✓ | ✓ | ✓ |
| Temperature scaling | ✓ | ✗ | ✓ |
| Model watermarking | ✓ | △ | △ |
| Knowledge distillation | △ | ✓ | ✗ |
| Weight noise injection | ✗ | ✓ | ✗ |

*Multiclass classification.* Each sample $i$ has a logit vector $Z_{out}^i$ of size $c$ (# classes). To reduce the confidence, we reduced the spread between highest and lowest logits for each sample, while maintaining the predicted class logit as the maximum. First, we add constraints to ensure that, for a given sample $i$, the logit of the predicted class $Z_{out}^{i,H}$ stays the highest within the logit vector (equation 9). Next, we add a constraint to minimize the spread between the highest logit $Z_{out}^{i,H}$ and lowest logit $Z_{out}^{i,L}$ (equation 10). For example, consider the $i$'th sample's logit vector as $-9, 7, 6, -4$, where the predicted class is 2. To maintain correct classification, the corresponding logit to the predicted class $Z_{out}^{i,2}$ must remain higher than other classes. To reduce model confidence, we enforce a reduction in the gap between $Z_{out}^{i,2}$ and the lowest logit, $Z_{out}^{i,1} = -9$.

$$Z_{out}^{i,H} \geq Z_{out}^{i,j} + \text{tol}, \quad \text{for all } j \in \{1, \ldots, c\} \setminus \{H\} \tag{9}$$

$$minimize \sum_{i=1}^{n} (Z_{out}^{i,H} - Z_{out}^{i,L}) \tag{10}$$

## 5.2 Empirical Results

Table 7 summarizes how our approach differs from existing techniques that modify NN behavior. Post-hoc calibration methods, e.g., temperature scaling (Guo et al., 2017) operate only at the output level, altering confidence estimates while preserving both training and test accuracy. Knowledge distillation (Hinton et al., 2015) alters model capacity and typically requires retraining and impacts training accuracy and generalization. Weight perturbation (Neelakantan et al., 2020) generally degrades training and test performance and is not designed for controlled behavior modification. Model watermarking (Adi et al., 2018) focuses on embedding verifiable signatures or trigger-based behaviors for ownership verification, instead of controlling or degrading test generalization under fixed training accuracy. In contrast, TAGD intentionally alters generalization behavior while preserving training accuracy, without retraining, distinguishing it from calibration and compression techniques by targeting generalization itself under explicit training accuracy constraints.

Table 8 presents the models' training accuracy and loss before and after applying TAGD. Notably, while the accuracy remains unchanged between the original model ($model_{init}$) and the modified model ($model_G$), the training loss exhibits a clear difference. This indicates that TAGD successfully perturbs the model to reduce its confidence without altering its classification outcomes. For all image datasets, the loss increased substantially after TAGD, while accuracy is unchanged. For instance, in *CIFAR10* the loss rose from 0.07 to 2.3, demonstrating a significant drop in confidence. Among the five tabular datasets, four showed an increase in loss. The only exception was the *higgs* dataset, where the loss decreased slightly from 0.52 to 0.51. This occurs because MILP constraints do not directly maximize loss, but instead minimize the models' prediction confidence. In such rare cases, this reduction in confidence does not translate to higher loss. We also observe that the increase in loss is generally larger for image datasets compared to tabular ones. This could be attributed to image datasets' multiclass classification, where logit vector contains multiple values (one for each class), giving the solver more degrees of freedom to alter the logits while keeping the predicted class unchanged. In contrast, tabular datasets involve binary classification, where the logit is essentially a single real number, leaving less room to adjust values without affecting the final prediction. As a result, loss increases are more limited for these models.

Table 8: TAGD: training and test accuracy (%) and loss

|  | Dataset | Training Set | | | | Test Set | | |
|---|---|---|---|---|---|---|---|---|
|  |  | $m_{init}$ | | $m_G$ | | $m_{init}$ | $m_G$ | $m_G - m_{init}$ |
|  |  | Accuracy | Loss | Accuracy | Loss | Accuracy | Accuracy | Accuracy |
| Image | CIFAR10 | 97.67 | 0.07 | 97.06 | 2.30 | 92.66 | 91.87 | **-0.79** |
|  | EMNIST | 96.66 | 0.09 | 96.66 | 3.26 | 94.60 | 93.96 | **-0.65** |
|  | FashionMNIST | 95.65 | 0.12 | 95.42 | 2.30 | 92.28 | 91.52 | **-0.76** |
|  | Food101 | 81.61 | 0.56 | 81.61 | 2.30 | 66.74 | 64.54 | **-2.19** |
|  | KMNIST | 99.95 | 0.00 | 99.95 | 2.30 | 97.58 | 97.58 | 0.00 |
|  | MNIST | 98.05 | 0.06 | 98.05 | 2.30 | 97.83 | 97.82 | **-0.01** |
|  | office31 | 98.62 | 0.08 | 98.62 | 3.36 | 75.60 | 19.18 | **-56.41** |
|  | SVHN | 95.56 | 0.16 | 95.56 | 2.30 | 94.18 | 94.04 | **-0.13** |
|  | USPS | 99.89 | 0.00 | 99.89 | 2.30 | 96.95 | 96.74 | **-0.21** |
| Tabular | Adult | 85.40 | 0.31 | 85.40 | 0.34 | 84.43 | 84.43 | 0.00 |
|  | higgs | 74.98 | 0.52 | 74.98 | 0.51 | 69.11 | 69.11 | 0.00 |
|  | GiveMeSomeCredit | 93.06 | 0.24 | 93.06 | 0.69 | 93.06 | 93.05 | **-0.01** |
|  | bank-marketing | 88.30 | 0.29 | 88.30 | 0.69 | 88.30 | 88.25 | **-0.06** |
|  | santander | 98.34 | 0.04 | 98.34 | 0.04 | 88.00 | 86.97 | **-1.03** |

Table 8 also shows a similar pattern for test accuracy. For image datasets, where logits span multiple classes, reducing the confidence of the correct class while keeping the prediction fixed leads to a drop in test accuracy. For example, *Food101* and *office31* show notable drops of 2.19 and 56.41% while *CIFAR10* and *SVHN* exhibit small but consistent declines. In contrast, the tabular datasets' models being binary classifiers, show almost no change (within 0.01), as TAGD has limited ability to impact these models without altering predictions.

## 6 Runtime Analysis

We illustrate the practical feasibility of MILP formulations by reporting solve times in different architectural and data-related settings. All timings reflect averages over 5 successful runs. We used a commodity desktop system, with an Intel Core i7-6950x CPU (3.0GHz, 10 cores) and 128GB RAM.

**Samples.** Figure 5 shows runtime on the *CIFAR10* and *MNIST* datasets when running (a) TAGD and (b) CMC, with varying subset sizes. TAGD runtime increases linearly, averaging 305 seconds for 10,000 *CIFAR10* samples. Since for CMC we used a subset size of 1,000, the runtime remains feasible in practice; even increasing the subset size to 20,000 results in an average solve time of 9,339 seconds ($\approx$ 156 minutes). Figure 5 (c) compares time to find the first feasible vs. optimal solution for CMC on *CIFAR10*. Notably, a solution is found within 6,246 seconds (105 minutes) for 20,000 samples, demonstrating CMC's practicality.

**Layers.** Figure 6 (a) shows that the total number of layers has almost no effect on MILP runtime. Although deeper networks take longer to train, the MILP operates only on the final layer, so the number of hidden layers or internal mechanisms does not influence the solver time.

**Nodes.** Figure 6 (b) shows the CMC runtime applied to the *CIFAR10* and *MNIST* datasets with varying final-layer widths from 16 to 1024. Since the width of the final layer directly determines the size of the MILP, the runtime increases accordingly; this scaling is smooth and predictable. Additionally, in most practical architectures, the final layer is relatively small, so this scaling factor is manageable.

**Classes.** We evaluated CMC on *CIFAR10* using 4, 6, 8, and 10 randomly selected classes. We observed a runtime linear in the number of classes (Figure 6 (c); each point is the average of five runs). The average time was 171 seconds for 4 classes and 492 seconds for 10 classes.

## 7 Related Work

**Post-hoc Model Generalization.** ROME (Meng et al., 2022) edits factual associations in language models by applying rank-one updates to transformer MLPs; while effective for precise single edits, this is

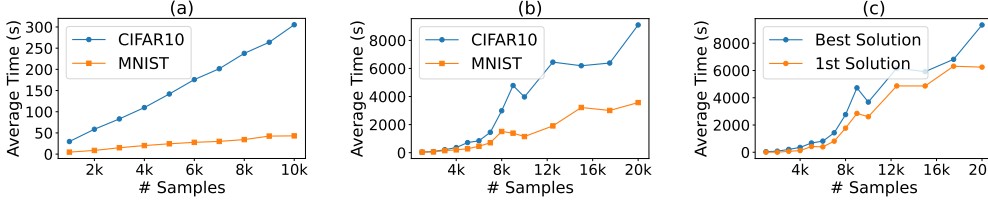

Figure 5: Solve time vs. sample size: (a) TAGD, (b) CMC, and (c) CMC 1st and optimal solution.

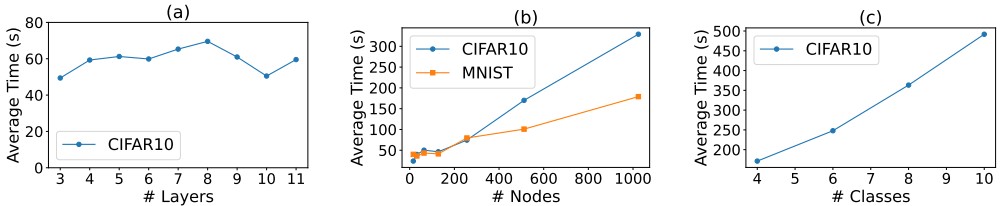

Figure 6: Solve time vs. number of: (a) layers, (b) nodes, and (c) classes

confined to NLP and offers no guarantees against unintended side effects. PMET (Li et al., 2024) edits transformer FFNs with minimal collateral impact, but lack formal guarantees and is confined to NLP. Model editing via gradient-based tuning or latent updates risks over-generalization and lacks locality (Mitchell et al., 2021). Blending task-specific weights via tangent-space arithmetic (Ortiz-Jimenez et al., 2023) lacks support for precise behavior edits. We address these limitations via MILP-inferred, verifiable label changes, regardless of architecture. To improve generalization, RCAD (Setlur et al., 2022) penalizes overconfidence on spurious features; while this may slightly lower training accuracy, it typically boosts test performance. Other methods fix fake patterns to improve generalization. PHATGOOSE (Muqeeth et al., 2024) inserts low-rank adapters trained with causal interventions to correct decision boundaries without full retraining. PCBM (Yuksekgonul et al., 2022) projects internal activations on a concept space to prune spurious features post hoc. While effective for robustness, these methods do not support precise, targeted behavior edits.

**Model Degradation and Obfuscation.** NNSplitter (Zhou et al., 2023) obfuscates weights via reinforcement learning; the model functions only with access to a secure set of "model secrets"; while effective for IP protection, such approaches lose predictive utility and produce incorrect outputs by design. Related efforts use hardware-dependent training, tying model functionality to a secure key (Chakraborty et al., 2020), or passport-based watermarking that degrades performance when unauthorized credentials are used (Fan et al., 2019). Fault injection (Liu et al., 2017) degrades NNs by flipping a small number of weight bits or injecting targeted faults, often leading to reduced accuracy. Applicability authorization (Wang et al., 2021) protects a model by restricting its utility to authorized data domains only, and degrading performance elsewhere. Restricting model generalization via adversarial augmentation (Qiao et al., 2020; Zhou et al., 2020) or entropy regularization (Zhao et al., 2020) can shape domain-specific behavior. Overconfidence can be reduced during training, to improve OOD detection or calibration (e.g., LogitNorm (Wei et al., 2022)) or by encouraging high-entropy output distributions (Pereyra et al., 2017).

## 8 Conclusion

We introduced a framework that treats neural network generalization as a constraint-solving problem via CMC. By using MILP to identify the minimal weight perturbations required to flip targeted training labels, we provide a principled way to move converged models out of overfitted regions. Beyond performance gains, the flexibility of our exact optimization approach allows for diverse behavioral edits, such as model obfuscation via TAGD. Experiments on image and tabular datasets show that our approach enables precise, verifiable interventions across architectures and tasks.

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
