# OpenReview forum: "Controlling Neural Network Generalization via Constraint-Guided Weight Transformations"
_TMLR — Under review for TMLR_

### Review · Reviewer_QYNd · 2026-07-01

**Summary Of Contributions:**

- This paper reformulates generalization improvement as a post-hoc, constraint-guided model editing problem: CMC finds the minimal final-layer weight/bias perturbation needed to flip the predicted labels of exactly m training points while leaving all other predictions unchanged, casting generalization as something actively steerable rather than a passive byproduct of training.

- Provides a concrete MILP encoding for both binary and multiclass settings, including misclassification flags, per-sample unsatisfied-indicator variables, the exactly-m constraint, the L1-minimization objective, and an optional constraint restricting flips to originally correctly-classified points.

- Shows the same MILP machinery can be inverted for controlled model degradation (TAGD), reducing test performance while holding training accuracy and predicted labels fixed, presented as a tool for model obfuscation.

- Includes a runtime analysis showing that restricting the MILP to the final layer keeps solve time independent of network depth and tractable on commodity hardware.

**Audience:**

Yes

**Audience Explanation:**

The paper sits at the intersection of model editing, generalization, and combinatorial optimization, and there is a plausible audience within TMLR for its core idea.

The dual use of the same framework for controlled degradation (TAGD) also touches an audience concerned with model obfuscation, IP protection, and access control, which is a smaller but active community.

**Broader Impact Concerns:**

The submission does not contain a Broader Impact Statement. TAGD component warrants one.

While the authors frame this positively as IP protection and access control, the same technique is dual-use: it provides a recipe for making a model appear to perform identically on benchmark data while silently underperforming on unseen inputs, without any visible change in labels. This has clear potential for deceptive use. For example, passing off a deliberately weakened model as the genuine one during evaluation or auditing, or evading capability assessments and the paper does not discuss safeguards, detectability, or the threat model under which such concealment would or would not be acceptable.

**Claims And Evidence:**

No

**Claims Explanation:**

The paper's central claim that flipping a small number of labels via minimal final-layer perturbations pushes the model out of sharp or overfitted regions of the loss landscape and thereby improves generalization is the mechanism the entire method rests on, yet it is asserted rather than demonstrated. No direct evidence is provided that the edited model actually occupies a flatter or less-overfitted region: there are no sharpness, curvature, or Hessian-based measurements before versus after the edit, and the loss-landscape framing is supported only by a schematic illustration (Figure 2) rather than by measurements on the models actually used in the experiments. The causal story connecting the edit to the observed accuracy changes is therefore not established.

The empirical evidence is also not convincing at the magnitude claimed. The reported gains are small and are presented without standard deviations, confidence intervals, or significance tests, so it is impossible to tell whether they exceed the run-to-run variance introduced by the additional 100-epoch retraining. This concern is sharpened by the absence of the key ablation: a control that resumes training for the same number of epochs without any perturbation. Without it, the possibility that the gains come from extra training rather than from CMC's edit is not ruled out, and several of the reported deltas sit at the same scale as the claimed improvements.

**Requested Changes:**

- The most important addition is a no-perturbation control. The paper resumes training for up to 100 additional epochs after the edit, so the reported gains cannot currently be attributed to CMC rather than to the extra training itself. I would need to see, for every dataset and training method, a baseline that resumes training for the same number of epochs from Model_init with no label flips, reported alongside the CMC numbers. Without this, the central claim that the edit improves generalization is unsupported.

-  All headline results need variance quantification.

- The mechanistic claim needs direct evidence or should be substantially softened. The paper repeatedly asserts that CMC moves the model out of sharp/overfitted regions, but provides no measurement on the actual models.

- An ablation on the subset size and on m beyond the two values tested (m=1, m=10) would clarify how sensitive the method is and whether the "one flip is best" finding is robust or an artifact of the search space.

- Reproducibility would benefit from reporting the fraction of the one-hour time limit actually consumed per dataset and the number of solver timeouts beyond the EMNIST/office31 C10 cases already noted, since the practicality argument rests on the solver reliably returning within budget.

---

### Review · Reviewer_jxjU · 2026-07-04

**Summary Of Contributions:**

The paper proposes Controlled Misclassification (CMC), a post-hoc model editing method that uses MILP to minimally perturb the final layer weights so that a small number of training samples intentionally change predicted labels. The authors argue that this controlled perturbation can move a converged model away from a poorly generalizing solution and improve test accuracy. They also introduce TAGD, a related MILP-based formulation that preserves training predictions while reducing generalization, framed as model obfuscation. The paper evaluates CMC on image and tabular datasets and reports modest test-accuracy gains.

**Strengths:**
1.  The paper introduces an interesting optimization perspective on post-hoc neural network editing.
2. Editing only the final layer keeps the method relatively simple and makes the optimization problem more tractable.
3. The TAGD formulation is conceptually interesting because it shows that the same constraint framework can also degrade generalization while preserving training behavior.

**Weaknesses:**
1. The central claim that CMC helps escape sharp or overfitted regions is not directly supported by evidence.
2. The experimental design does not clearly separate the effect of CMC from extra training, validation selection, or random perturbation.
3. The L1 minimal-perturbation objective is not theoretically justified as a mechanism for improving generalization.
4. Scalability is a concern because the method depends on a MILP solver and becomes expensive as the subset size or final-layer dimension grows.
5. The writing overstates the contribution, such as “controlling generalization” is stronger than what the current results demonstrate.

**Audience:**

Yes

**Audience Explanation:**

The paper’s use of MILP for post-hoc neural network weight perturbation is an interesting idea for readers in optimization, model editing, and neural network control. However, the findings are not broadly compelling yet because the empirical gains are small, the mechanism is not clearly validated, and the method is limited to final-layer edits on standard datasets.

**Broader Impact Concerns:**

The paper should include a broader impact discussion. CMC and TAGD are post-hoc model editing methods that can alter model behavior without changing architecture or training labels, so the paper should discuss risks around deceptive model release, benchmark manipulation, and reduced transparency in safety-critical applications. The authors should also clarify appropriate use cases, limitations, and safeguards

**Claims And Evidence:**

No

**Claims Explanation:**

1. The main empirical gains are small and there is no clear statistical evidence is provided. So say that, random seed variation could easily explain part of the improvement, not from the proposed methods.
2.  CMC includes additional retraining after perturbation, so the paper needs a strict control, such as continue training the original model for the same extra budget without CMC. Without this, it is unclear whether improvement comes from the MILP perturbation or simply from more training.
3. TAGD shows the MILP can degrade test accuracy while preserving training predictions, but this is almost a separate problem to me. It does not strongly support the claim that CMC improves generalization.

**Requested Changes:**

1. Add a fair continued training baseline. It is my major concern. For example, compare CMC against simply continuing training the original model for the same number of extra epochs.
2. Separate the effect of MILP from generic perturbation. Add baselines such as random final-layer perturbation, adversarial/randomly selected label flips, and small L1/L2 perturbations followed by retraining.
3. The paper studies some settings, but a more systematic sensitivity analysis would help explain when CMC works or fails, such as  add ablations on the number of flipped samples and subset size.
4. The paper should frame CMC more accurately as a post-hoc final-layer perturbation that sometimes improves generalization.
5. TAGD is interesting but feels separate from the main generalization claim. The paper should either connect it more clearly to the main contribution or move it to an auxiliary section.

---

### Review · Reviewer_Ew9s · 2026-07-19

**Summary Of Contributions:**

## Summary
The paper introduces Controlled Misclassification (CMC), a post-hoc model editing framework that improves neural network generalization by using mixed-integer linear programming (MILP) to make minimal, constraint-guided perturbations to the final layer of a trained model. By deliberately flipping the predicted labels of a small subset of training samples and then resuming training, CMC pushes the model toward better-generalizing solutions: achieving up to 2.8% test accuracy improvement across diverse image and tabular benchmarks, and complementing existing techniques such as SAM and AWP. The authors further demonstrate the framework's versatility through TAGD, which uses the same MILP formulation to degrade generalization while preserving training accuracy, enabling model obfuscation. Together, these contributions reframe generalization as an constraint-satisfaction problem rather than a passive outcome of training.

## Strengths
- [S1] **Complementarity to existing methods.** CMC can work on top of existing generalization methods (SAM, AWP, RWP) rather than competing with them. This is a strong property in practicality, which positions the method as a low-risk addition to current pipelines.
- [S2] **Verifiable post-hoc edits to the model.** Unlike gradient-based fine-tuning or other perturbation approaches, the MILP formulation provides transparency about what changes: you can tell precisely which samples are flipped and can verify that the perturbation is minimal in an L1 sense. This interpretability seems like a strength for trustworthiness and debugging.

## Weaknesses
- [W1] **Unconvincing motivation for TAGD.** The stated goal of TAGD is to "retain control over the original model while providing users a functional version," yet the method deliberately degrades test-set performance. A model that classifies training data correctly but fails on unseen inputs is not meaningfully "functional" for the users. The disconnect between the claimed use case (model obfuscation ) and the actual outcome (a model that simply performs poorly in deployment) undermines the practicality of this contribution.
- [W2] **Insufficient justification for CMC via MILP.** The key value proposition of the constrained formulation is that the MILP solver selects which samples to flip in a principled way. However, comparing the constrained variants (C1, C10) against the arbitrary variants (A1, A10) in Table 2 doesn't seem to reveal statistically significant benefits. If randomly selecting samples to flip yields comparable gains, the computational cost of solving the MILP is difficult to justify, and the contribution reduces to the observation that any small perturbation followed by retraining can help.
- [W3] **Difficulty isolating CMC's effect in Table 4.** The best results are reported as combinations (e.g., SAM+CMC, AWP+CMC), but the paper does not adequately disentangle how much of the improvement stems from CMC versus the base method. A clearer ablation on CMC's marginal contribution with controlled comparisons would be more informative
- [W4] **No tangible justification for why this works.** The intuition that flipping labels pushes the model out of sharp minima, is borrowed loosely from the sharpness-aware literature, but the paper never formally verifies that perturbed models actually land in flatter regions. There are no loss landscape visualizations or sharpness measurements to support the claimed mechanism.
- [W5] **Single-run reporting and small effect sizes.** Many reported gains in performance are very modest, and results appear to come from single runs without confidence intervals or significance tests. For instance, many numbers in Table 2 are below 1%, and is shown without error bars, making it impossible to tell if any of the differences are meaningful.
- [W6] **Limited exploration of the hyperparameter $m$.** Only misclassifying 1 or 10 images are tested, without any systematic study of how performance scales with the number of flipped samples, which limits practical guidance.

**Audience:**

Yes

**Audience Explanation:**

The paper offers a novel framing of generalization as a constraint-satisfaction problem, which is conceptually an interesting form of regularization and optimizer design. Researchers working on post-hoc model editing, loss landscape geometry, or MILP applications in machine learning would likely find the approach interesting, even if the empirical evidence is not yet fully convincing.

**Broader Impact Concerns:**

No concerns on any ethical implications of the work.

**Claims And Evidence:**

No

**Claims Explanation:**

In Table 2, the constrained variants (C1, C10) show no clear advantage over arbitrary selection (A1, A10), undermining the core justification for the MILP formulation [W2]. The reported gains are also often below 1% from apparent single runs without confidence intervals, making it difficult to distinguish signal from noise [W5]. The mechanistic claim about escaping sharp minima remains unverified [W4]. Altogether, these gaps leave the claims plausible but inadequately supported. Please refer to the block above for further details.

**Requested Changes:**

- [W1] Clarify/revise the TAGD motivation. Either provide a more convincing use case where a model that fails on test data is genuinely "functional" for users, or reframe TAGD's contribution more honestly as a demonstration of the framework's flexibility rather than a practical tool. Some literature in model obfuscation might help as well.
- [W2,W5] Demonstrate that MILP-guided selection outperforms arbitrary selection. Statistical significance tests (e.g., confidence intervals across multiple seeds) comparing C1/C10 against A1/A10 would help since without this, the core motivation for solving the MILP is not justified.
- [W3] Isolate CMC's marginal effect more clearly in Table 4. It would be better to report the improvement delta of adding CMC to each base method in a way that controls for any confounds, rather than only reporting combined final accuracies in ranks.
- [W4] Provide empirical evidence for the claimed mechanism. Loss landscape visualizations or sharpness measurements (e.g., comparing the Hessian trace before and after CMC perturbation) could help verify the intuition that CMC pushes models toward flatter minima.
- [W6] Experiment with wider range of flip counts m. Testing $m$ values beyond 1 and 10 (e.g., 2, 5, 20, 50) could help characterize how performance scales, ideally providing guidance on how users should select the hyperparameter.